# Culturally Responsive Middle Leadership for Equitable Student Outcomes

Camilla Highfield *[ID], Melinda Webber and Rachel Woods

Faculty of Education and Social Work, University of Auckland, Private Bag 92019, Auckland 1142, New Zealand; m.webber@auckland.ac.nz (M.W.); rachel.woods@auckland.ac.nz (R.W.)
* Correspondence: c.highfield@auckland.ac.nz

**Abstract:** Middle leaders are important conduits for school policy and are often required to lead, champion and monitor change initiatives in their departments. This paper examines secondary school middle leaders' self-reported theories, promotion and leadership of culturally responsive teaching approaches for student equity. The study participants (*n* = 170) are curriculum leaders in state secondary schools throughout New Zealand, serving a range of diverse populations, including Indigenous Māori students. The quantitative and qualitative data for this project were thematically analysed to inquire into middle leaders' articulation of their culturally responsive theories, practices, and pedagogical and curriculum leadership to assist members of their departments in supporting Māori student educational success. The study found that although middle leaders could clearly articulate culturally responsive, sustaining and revitalising pedagogies and leadership in alignment with government policies and expectations, they were less clear about the measures they used to evidence these practices. In addition, less than 30% of the middle leader participants mentioned academic achievement as an element of Māori student success, and few mentioned the importance of systematic monitoring of achievement data, or using them to support better learning outcomes for Māori students (184).

**Keywords:** secondary schools; curriculum middle leaders; culturally responsive; indigenous

## 1. Introduction

This study reports evidence of middle leaders' perceptions of their culturally responsive pedagogy (CRP) through descriptive analysis of quantitative data and thematic analysis of qualitative data to understand the extent to which they believe their CRP is effective in supporting Indigenous Māori (hereby referred to as Māori) students to gain success at school. The self-reported beliefs of middle leaders were then compared to the university entrance results of Māori students attending the school in the same year the middle leader data were collected. Middle school curriculum leaders play a critical role in the school hierarchy, which often includes responsibility for pedagogical leadership, the professional development of teachers in their department, oversight of curriculum development, and the close tracking of student achievement [1]. The research reported in this paper utilises intersectionality [2] as a tool to examine the potential relationship between the beliefs and CRP practices of middle leaders in secondary schools and the academic success of Māori students studying within the curriculum departments they lead. The guiding research questions for this study are "How do secondary middle leaders articulate and lead culturally responsive departmental practices?" and "Do the academic results of Māori students at whole-school level vary in relation to the self-reported CRP practices of middle leaders in that school?" Intersectionality is a framework that has been used by researchers to understand the colonising relationship between Indigenous people and the nation-state [3]—in this case, the education system and school. The core ideas can be applied to educational leadership [4] and advance a larger goal of increasing the

commitment of leaders in schools to focus on improved outcomes for Indigenous students. Crenshaw identified three forms of intersectional analysis: structural, political and representational [5]. This paper highlights the structural influences of government, school policy, principal management and leadership in their impact on middle leader beliefs and their practices within secondary schools. The extent to which school principals are able to modify government policy and develop alternative approaches in order to progress value-led approaches to school leadership and management [6] also contributes to the notions of intersectionality that are evident in this juxtaposition of policy, practice and Māori student academic outcomes. Connolly, James and Fertig explain how educational management entails carrying responsibility for the effective functioning of a school in which others participate [7]. Management requires influencing others to achieve goals and necessitate actions of some kind. When those carrying a delegated responsibility act in relation to that responsibility, they influence and are therefore leading. Educational leadership is ideally undertaken responsibly and is critical if the welfare and academic outcomes of Māori students are to be prioritized. Connolly, James and Fertig [7] assert that educational management entails delegation which in the school system means middle leaders are assigned and carry the responsibility for policy enactment, which, in New Zealand, is focused on raising achievement for equity groups [8].

The survey questions used in this study were designed to elicit middle leader understanding, articulation and reporting of actions to support the educational success of Māori students. Given middle leaders have considerable influence on the outcomes of students studying within the curriculum departments they lead, understanding the steps they take to address inequities is critical. There is limited research in the middle leadership literature that examines the influence of culturally responsive pedagogy and the academic outcomes of Māori students in the secondary school context [9]. Capper [2] argued that intersectionality within the field of equity leadership is worth pursuing, and in this study, we provide evidence of the instructional practices of middle leaders and the ways in which they do and do not articulate the culturally responsive pedagogy they espouse in support of educationally powerful outcomes for Māori students [10].

*The New Zealand Context*

New Zealand has one of the most highly devolved education systems in the world [11]. Individual secondary schools are governed by a Board of Trustees, elected by the parent group and managed by the principal and staff [12]. Boards are responsible for the management of the school and are legally responsible for ensuring schools operate within the framework of government regulations [13]. Moreover, the 2020 Education and Training Act [14] stipulates that New Zealand's founding constitutional document, Te Tiriti o Waitangi, obliges schools, as Government organisations, to ensure that they are bringing Te Tiriti o Waitangi into effect. The 2020 Act stipulates that schools must give effect to Te Tiriti by working to ensure their plans, policies and local curriculum reflect local tikanga Māori (protocols), mātauranga Māori (knowledge/wisdom) and te ao Māori (worldviews) [14]. Principals are responsible for the culturally responsive leadership of curriculum, continual improvement, caring for students, modelling the school values, maintaining integrity and problem solving [15].

The organisation of secondary schools into subject departments is a common feature of schools serving the education needs of 13–18-year-olds in many Western countries, despite the diversity in the size, location, vision and governance style of schools [16]. In New Zealand, secondary schools structure themselves with standardised departmental labels which divide teachers and courses along curriculum-related lines. The term *middle leaders* in this article refers to those individuals with responsibility for curriculum leadership, often referred to in New Zealand secondary schools as a head of department. Middle leaders are positioned in the centre of the school hierarchy, beneath senior leaders, and have responsibility for leading teachers [17]. Glover et al. referred to the middle leader role as involving both upward communication of departmental opinion into the wider school hierarchy and

downward communication to teachers and mediation of external demands [18]. The reason for the focus on middle leaders in this study is their unique position in the management and translation of schoolwide goals and their practical leadership of change initiatives with and alongside the teachers in their department.

New Zealand has a similar history of colonisation to countries such as Canada, Australia and the United States where successive governments asserted domination and control over Indigenous people [3]. The British colonisers in New Zealand had an ongoing assimilation strategy, attempting to dismantle and erase Māori society, language and culture and replace it through religious, political and economic conversion [3]. However, during the early period of colonisation in the 1880s, Māori were actively participating, and in some cases leading, in the changing economic and technological landscape. Māori uptake of new technologies, literacy and trade opportunities challenged British perceptions of their own cultural superiority and inherent right to rule Indigenous populations [19]. Moreover, Hoskins et al. contended that Māori have always been (and remain) key actors in New Zealand schooling, responding flexibly and strategically to achieve educational success, and caution us against "rendering Māori communities [as] always-already victims, devoid of authority, resistance, strategic engagement and mana" [20] (p. 150). Similarly, Hetaraka argued that teachers need to instead reflect on the ways political histories and educational policies have influenced the education system in order to question their own theories, practices and conscious and unconscious biases [19]. Teachers must be continuously alert to the ongoing ways the colonial education system imposes low teacher expectations of Māori students, dismisses Māori knowledge as inappropriate for inclusion in the curriculum, and engages in sustained deficit theorising and negative stereotyping about Māori students and their families. Attention to the ways these actions result in Māori disengagement from the education system is vital [10].

The variable education success of Māori students has been attributed to the uneven quality of teaching experienced [21]. Researchers have found that teachers' stereotypical views attributed the poor academic success of Māori students to their families being uneducated, their personal dispositions and their academic capabilities [22–24]. These deficit views create negative and problematic student–teacher relationships, lower teacher expectations, and lessen the amount of agency teachers feel in terms of their power to improve outcomes for Māori students [25]. Over the last 10 years, the New Zealand government has developed policies, guidelines, resources and professional development opportunities for New Zealand teachers and leaders to shift the outcomes for students who are underserved by the education system, particularly Māori students. The central policy statement *Ka Hikitia* has attempted to support decolonisation efforts through a vision of Māori students living and learning *as* Māori [26]. Key outcomes of the policy relate to Māori students achieving excellent education outcomes, developing a sense of belonging within and across the education system, building on their cultural strengths and assets, and encouraging productive partnerships between Māori students, their families and schools [26]. However, there is little research examining the extent to which these policy directives have impacted the beliefs, practices and leadership of middle leaders.

## 2. Middle Leaders Championing Change

Middle leaders are conduits for school policy and are often required to lead and champion change initiatives with the teachers in their departments. Educational research has recognised the important role of middle leaders in embracing the curriculum and pedagogy required to promote student academic achievement [27]. This critical position within the school hierarchy includes responsibility for pedagogical leadership within their departments, including a mandate to ensure that both professional development and curriculum development occur [28]. Middle leaders in secondary schools are required to enact both instructional and transformational leadership practices [29]. This means they play a pivotal role in securing better learning outcomes for students [30] while modelling integrity, developing trust within their team, setting clear goals, encouraging high expectations, and

moving teachers beyond their immediate self-interests [31]. These leadership practices need to be balanced alongside the demanding administration and management tasks that are included in the middle leader role description due to the acute challenges and pressures from school-level leaders and the teachers who report to them [30].

This study examined the congruence between what was theoretically espoused by middle leaders when they described their beliefs and leadership practices for Māori student success and what they described as evidence of their efforts. Argyris and Schon referred to this as "theories of action" and "theories in use" [32] (p. xxviii). Theory is often conceived of as an abstract idea or phenomenon, but leadership practice involves an action component that goes beyond the abstraction of theory. In this sense, theory represents knowledge, while practice is the application of that knowledge [33]. Teachers' theories and beliefs regarding their leadership and teaching practices, such as their perceptions of students' abilities, what knowledge is of most worth, and the value of certain teaching techniques and pedagogical principles, have a profound impact on their day-to-day decision making [34]. These beliefs drive teachers' actions and may be used to justify or validate their chosen pedagogies. However, many teachers are unaware of their assumptions, theories or educational beliefs and sometimes adopt ideas that have the ring of fashionable rhetoric or that coincide with the expectations of certain others, such as a principal they admire [35]. Publicly they may espouse ideas and assume their classroom behaviours are guided by these ideas, but privately or even unknowingly they may believe something else that actually governs their classroom behaviour [34].

## 3. Culturally Responsive Pedagogy

CRP is grounded in educational philosophies, practices and policies that enable an inclusive schooling environment for students and families from ethnically and culturally diverse backgrounds. Ladson-Billings first coined the term *culturally relevant pedagogy* and proposed that it rested on three key propositions: (a) students must experience academic success; (b) students must develop and/or maintain cultural competence; and (c) students must develop a critical consciousness through which they challenge the status quo of the social order [36,37]. Later, Gay offered a similar term, contending that teachers who engaged in culturally responsive teaching must have a sound knowledge base about cultural diversity; know how to integrate the cultural characteristics, experiences and perspectives of ethnically diverse students in the curriculum; demonstrate culturally nuanced care for culturally diverse students; develop inclusive learning communities; use effective cross-cultural communication strategies; and respond to diversity in the delivery of instruction [38,39].

In the New Zealand context, where Māori students and their families remain underserved, CRP necessitates pedagogical and cultural change beyond the school gates. Teachers' practice must strengthen the Māori language, identity and culture, and build and sustain enduring school–community partnerships [40,41]; advance co-constructed localised curricula [42,43]; and ensure an education free from racism, stigma and discrimination [10] where there are high expectations for academic success. CRP requires middle leaders, and the teachers who report to them, to engage in responsive and timely strategies to meet the diverse needs of Māori students, the multiple demands of schools, and government and school policies and requirements.

Government policy and professional development programmes in New Zealand have encouraged middle leaders to advocate for and advance CRP in their departments [44–47]. CRP practices that advance Māori student achievement require middle leaders to strategically and collaboratively identify, implement and lead departmental strategies that strengthen teaching and learning [48]. Johnson contended that culturally responsive middle leaders should use their influence (and co-ordinate the influence of others) to "work against the grain" [49] (p. 162) of educational bureaucracies, consciously linking schools and community improvement efforts. Johnson also argued that these experiences help leaders to develop their own critical consciousness [49]. Bishop contended that CRP involves

teachers and leaders demonstrating three broad capabilities: creating family-like contexts for learning, interacting with students in ways that promote learning, and monitoring students' progress and the impact of their teaching on how well students are able to self-manage and take ownership of their learning [21]. Monitoring the instructional practices of teachers to gauge their effectiveness for students within curriculum departments is a key role of middle leaders.

Middle leaders need to ensure that students in their departments are provided with every opportunity to be effectively taught and guided towards goals that will support the achievement of their academic aspirations. Franco et al. argued that when teachers balance the provision of warmth, care and emotional support with high expectations and structure for academic and social achievement, there is a link to engagement in classroom instruction and higher scores in standardised tests [50]. This is important because students who leave secondary school with minimal qualifications are increasingly disadvantaged in today's society because of the increasing requirements for knowledgeable, creative and technologically fluent employees within the global economy [51]. Students who fail to complete secondary school are more likely than graduates to be unemployed or underemployed as adults [52]. Johnston et al. [53] argued that students' future pathways are related to their teachers' expectations, especially in cases where students are from low socioeconomic and disadvantaged backgrounds, or identify as an ethnic minority [54,55]. Furthermore, lower teachers' expectations are more strongly associated with the academic performance of students from disadvantaged backgrounds than students from more privileged circumstances [56,57]. In New Zealand, students who identify as Māori can also be vulnerable and therefore are more likely to experience pronounced teacher expectation effects. In this study, we investigate the self-reported CRP practices of middle leaders at the school level and compare these aggregated results with the whole-school UE academic results of Māori students in order to understand the extent to which CRP practices of middle leaders could be associated with academic achievement of the Indigenous equity group.

## 4. Methodology

The results reported in this paper include quantitative and qualitative data collected using the Kia Tu Rangatira Ai Survey [58], which was distributed extensively throughout New Zealand primary and secondary schools to investigate the factors that support student engagement from the perspective of students, families and teachers/school leaders. Ethical approval for the study was granted by the University of Auckland Human Ethics Committee (UAHPEC Approval Number: 021775) and included voluntary informed consent. Participant confidentiality and anonymity were preserved in the collection, analysis and storage of data. The data reported in this paper feature the perspectives of 170 middle leaders employed in 12 secondary schools between November 2018 and August 2020. Eleven schools were situated in the North Island of New Zealand and one school was situated in the South Island. The schools were located in a range of geographic areas, with 28% of participants working in schools in metropolitan areas, 58% working in schools in large regional centres, 4% in schools situated in small regional centres and 11% in schools in rural areas [59]. The schools are a mix of coeducational schools (8) and single-sex schools (4). Most schools are fully funded by the government, although two schools are state-integrated schools where funding is received via a combination of government money and parental fees. Prior to 2023, the Ministry of Education (MOE) used a decile rating system from 1 to 10 to measure the socioeconomic status of students attending a school. The lower a school's decile rating, the more government money it received to provide extra resources to support its students' learning needs [60] (see Table 1).

Middle leader responses were separated from other teacher/leader responses when participants reported their role as being "head of department/faculty" (HOD/F), "assistant HOD/F", "teacher in charge" or "leader of learning". Middle leader respondents were drawn from a broad range of curriculum areas, including arts (12%), English (8%), health and physical education (9%), languages (4%), mathematics (4%), sciences (12%), social

sciences (8%), technology (14%) and inclusive education (4%). The remainder of middle leader survey respondents did not specify their curriculum area (25%) (See Table 2).

**Table 1.** Profile of Schools, 2020.

| School | Decile | % of Māori | # of ML Respondents | UE Rate |
|---|---|---|---|---|
| 1 | 3 | 54% | 6 | 26.7% |
| 2 | 5 | 20% | 12 | 33% |
| 3 | 7 | 18% | 34 | 34.7% |
| 4 | 8 | 14% | 13 | 66.7% |
| 5 | 6 | 30% | 18 | 31.7% |
| 6 | 6 | 27% | 17 | 61.3% |
| 7 | 5 | 39% | 6 | 22.2% |
| 8 | 4 | 53% | 18 | 42.3% |
| 9 | 4 | 55% | 7 | 23.1% |
| 10 | 7 | 18% | 10 | 28.6% |
| 11 | 5 | 40% | 9 | 26.8% |
| 12 | 5 | 40% | 20 | 37.3% |

Note: ML = middle leader; UE = university entrance.

**Table 2.** Demographic Data of Participants.

| | | Gender | | | Main Ethnicity | | | | | Years Spent Teaching | | | |
|---|---|---|---|---|---|---|---|---|---|---|---|---|---|
| | | M | F | Other | Pākehā | Māori | Pasifika [1] | Asian | Other | 3–8 | 9–14 | 15–20 | 21+ |
| | | % | % | % | % | % | % | % | % | % | % | % | % |
| Gender | M | 41.8 (71) | | | 39.2 (58) | 54.5 (6) | 66.7 (2) | 50 (2) | 75 (3) | 32.3 (10) | 48.5 (16) | 39.1 (18) | 45 (27) |
| | F | | 57 (97) | | 59.4 (88) | 45.5 (5) | 33.3 (1) | 50 (2) | 25 (1) | 67.7 (21) | 48.5 (16) | 60.9 (28) | 53.3 (32) |
| | Other | | | 1.2 (2) | 1.4 (2) | | | | | | | 3 (1) | 1.7 (1) |
| Main ethnicity | Pākehā | 81.7 (58) | 90.7 (88) | 100 (2) | 87 (148) | | | | | 83.9 (26) | 87.9 (29) | 86.9 (40) | 88.4 (53) |
| | Māori | 8.5 (6) | 5.2 (5) | | | 6.4 (11) | | | | 12.9 (4) | 6.1 (2) | 6.5 (3) | 3.3 (2) |
| | Pasifika | 2.8 (2) | 1 (1) | | | | 1.8 (3) | | | | | 2.2 (1) | 3.3 (2) |
| | Asian | 2.8 (2) | 2.1 (2) | | | | | 2.4 (4) | | | 3 (1) | 2.2 (1) | 3.3 (2) |
| | Other | 4.2 (3) | 1 (1) | | | | | | 2.4 (4) | 3.2 (1) | 3 (1) | 2.2 (1) | 1.7 (1) |
| Years spent teaching | 3–8 | 14.1 (10) | 21.6 (21) | | 17.6 (26) | 36.4 (4) | | | 25 (1) | 18.2 (31) | | | |
| | 9–14 | 22.5 (16) | 16.5 (16) | 50 (1) | 19.6 (29) | 18.2 (2) | | 25 (1) | 25 (1) | | 19.4 (33) | | |
| | 15–20 | 25.4 (18) | 28.9 (28) | | 27 (40) | 27.2 (3) | 33.3 (1) | 25 (1) | 25 (1) | | | 27.1 (46) | |
| | 21+ | 38 (27) | 33 (32) | 50 (1) | 35.8 (53) | 18.2 (2) | 66.7 (2) | 50 (2) | 25 (1) | | | | 35.3 (60) |

Note: Values in brackets show participant numbers. [1] Umbrella term unique to New Zealand used to describe the ethnicity of an individual who has migrated from a Pacific Island and their descendants. https://tapasa.tki.org.nz/about/tapasa/pacific-and-pasifika-terminology/ (accessed on 13 March 2024).

*Data Analysis*

The teacher survey contained 11 Likert questions using a 5-point response framework (*not at all true*, *a little bit true*, *somewhat true*, *mostly true* and *very true*) to measure the overall perception of middle leaders regarding their culturally responsive classroom practice, and then their perceptions of their own practice with regard to 10 separate elements of CRP (including questions about care, safety, respect, academic achievement, relationships, inclusion of family and knowledge of local history). The additional quantitative question asked middle leaders to compare their culturally responsive practice with other teachers in their school. The data were downloaded from the online survey tool and sorted using Microsoft Excel.

The survey contained six qualitative questions designed to elicit teachers' and school leaders' perceptions of teaching practices that support Māori student success. Thematic analysis was undertaken of middle leader responses to three questions in the survey:

1. How do you define Māori student success?
2. What teaching practices make a positive difference for Māori students at your school? What works?
3. What evidence do you have that these practices have made a positive difference? How do you know they work?

Two research assistants coded the data using an inductive approach to thematic analysis [61]. The researchers initially coded the first 30 responses to each qualitative question, drawing codes from the data. If middle leader responses included more than one comment they were placed into multiple codes. Once the emergent themes and codes were fully understood and agreed upon by both research assistants, coding was completed on all 170 survey responses separately with researchers meeting regularly to ensure coding agreeance and interrater reliability. The codes were then reviewed by the wider research team and refined to ensure the data reflected the diversity of middle leader responses.

## 5. Results

In respect of middle leader self-efficacy regarding their own culturally responsive practice, the overall responses to the Likert-scale questions in the survey were very positive (see Table 3).

**Table 3.** Middle Leader Culturally Responsive Practices—By % Who Responded Mostly True or Very True.

| Survey Statement | % Middle Leaders (n = 170) |
| --- | --- |
| *I ensure Māori students feel strong and safe in their cultural identity* | 84 |
| *I know when Māori students are achieving* | 91 |
| *Māori whānau (families) are made to feel welcome in my classroom* | 77 |
| *I treat Māori whānau (families) and Māori culture with respect* | 98 |
| *Māori whānau (families) are provided with opportunities to share their knowledge and experiences in my classroom* | 56 |
| *Māori students have multiple opportunities to succeed in my classroom* | 92 |
| *In my classroom, I know my Māori students and they know me* | 88 |
| *In my classroom, I respect the Māori students and they respect me* | 95 |
| *In my classroom, Māori students feel cared for* | 94 |
| *I know and teach the Māori history associated with where my school is based (e.g., hapū/iwi [tribal/local Māori] history)* | 31 |

Of the 10 Likert-scale questions, 5 showed over 90% of middle leaders believed it was *mostly* or *very true* that they used culturally responsive practices in the classroom. Middle leaders scored themselves particularly highly with regard to questions that measured their respect for Māori culture and relationships with students. Middle leaders were less confident (56%) about providing Māori whānau (families) with opportunities to share their knowledge in the classroom, and responses showed that just 31% felt confident about teaching the Māori history associated with where their school is based. These findings provide an indication of generally how few middle leaders provide opportunities for Māori families to be involved in teaching Māori students, and how even fewer middle leaders have the knowledge and confidence to teach key aspects of local Māori history to the students in their classrooms, despite reporting a positive attitude to Māori students.

Figure 1 provides an analysis of middle leaders' overall self-efficacy regarding culturally responsive pedagogy and compares middle leader responses to Māori student achievement in each of the schools in order to understand patterns and relationships in the data. Middle leaders responded to the question, "Relative to other teachers in your school, how culturally responsive to students' needs do you think you are?" Figure 1 shows that most middle leaders describe themselves as being average or above average in their understanding and enactment of culturally responsive approaches for students, but there is considerable variability in self-reported CRP practices within and across schools.

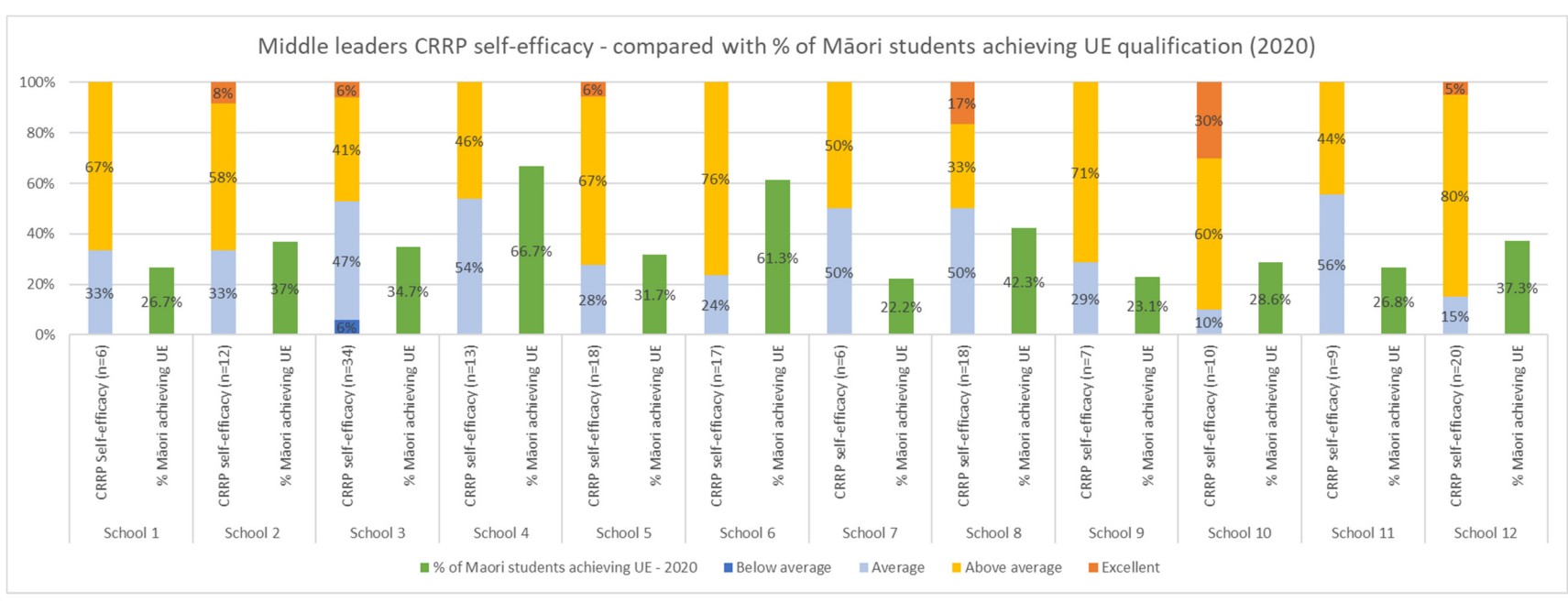

**Figure 1.** Middle Leaders' Self-Efficacy Regarding CRP Compared to Māori Student Academic Achievement of UE.

Figure 1 compares middle leaders' self-efficacy regarding CRP in relation to the percentage of Māori students on the roll gaining UE (https://www2.nzqa.govt.nz/ncea/understanding-secondary-quals/university-entrance/ (accessed on 13 March 2024) in their schools in 2020 when the data collection was undertaken [62]. In New Zealand, the highest qualification that can be gained at school is the University Entrance award (UE), which is the minimum required for direct entry to university from a New Zealand secondary school. To be awarded UE, a student needs to attain the National Certificate of Educational Achievement (NCEA) Level 3, with a minimum of 14 credits at Level 3 in each of three UE-approved subjects and the required credits in literacy and numeracy [62]. As a context for comparison, the data at the national level in 2020 revealed just 34% of Māori achieved UE compared with 64% of Asian and 59% of Pākehā students [63].

Figure 1 illustrates the stark variability in Māori student UE rates in the study schools ranging from 22% to 67%, with two-thirds (8/12) of the secondary schools in the study reporting Māori UE rates less than the national average. The school with the highest decile (eight) and the lowest percentage of Māori students on the school roll (14%) had the highest Māori UE pass rate (67%). The next two secondary schools (with decile ratings of seven) had school rolls with 18% Māori students and pass rates of 29% and 35%, respectively. The school with the lowest UE pass rate (22%) had a mid-decile rating (five) and 39% of Māori students on the roll. The three lowest decile schools (3, 3 and 4) had pass rates of 23%, 27% and 42%, respectively, and Māori rolls between 53 and 55%.

There was a range of diverse responses when middle leaders were asked Question 2, about the teaching practices that made a positive difference for Māori students at their school (Table 4). Middle leaders had clarity regarding the range of teaching practices that support Māori students to feel affirmed and acknowledged in the classroom. The greatest proportion of comments (59%) explained the specific actions and pedagogies within the classroom that teachers could enact. These comments reflected phrases synonymous with current research and government policy directives regarding the specific teaching practices known to make a positive difference in education outcomes for Māori students. Many middle leaders commented on the need to establish and maintain positive relationships with students, through taking an interest in their students' lives or developing positive relationships. A further 32% of comments specifically mentioned the positive affirmation of the culture and language background of Māori students as an important aspect of classroom practice. Just 4% of middle leaders commented on the importance of connecting with students' whānau in a positive way, and 5% of respondents did not answer this question.

In the qualitative section of the survey, middle leaders were initially asked how they define Māori student success (see Table 5). There was a total of 321 responses from 170 participants, which indicated many middle leaders had clear ideas about the types of evidence of success they were aspiring to support. Nearly half of all comments (46.6%) specifically related to achieving success, including the importance of academic success (24%) and Māori students meeting their goals in areas that were important to them (19.6%). Additionally, 37.4% of all comments related to students being engaged in learning or enjoying school as defining their success.

The third question asked middle leaders about evidence of where their culturally responsive practices had made a positive difference in Māori students' education success (see Table 6). The responses that referenced evidence of success with tangible data were those that described the improved assessment scores for students (20.4%) and increased attendance at school (4.2%). Nearly half the participants (47.1%) commented on positive student engagement as being evidenced by students having an encouraging attitude towards learning, showing pride in their culture, and affirmative student feedback. Relationships between teachers and students and with students' whānau were also mentioned by 15.6% of the respondents.

**Table 4.** Middle Leader Perspectives of the Teaching Practices That Make a Positive Difference for Māori Student Learning.

| Theme | Coded Responses | % of ML Comments (n = 170) |
|---|---|---|
| Inclusion of culture and language 32% | Affirming/acknowledging culture | 11% |
| | CRP—connect culture to learning | 12% |
| | Learn and use te reo to normalise use in the classroom | 5% |
| | Pronunciation of te reo and names | 4% |
| Specific teacher actions/pedagogy 59% | Create positive relationships/get to know student/take an interest | 23% |
| | Differentiated learning/assessment | 11% |
| | Be consistent/positive feedback/caring/patient/listen/use humour | 8% |
| | High expectations | 5% |
| | Respect student/believe in student/celebrate success | 5% |
| | One-on-one assistance | 3% |
| | Build a safe/inclusive learning environment for all/sense of belonging/feeling comfortable | 2% |
| | Set clear boundaries | 2% |
| Whānau involvement 4% | Whānau connection/partnership | 4% |
| No response 5% | Blank or "I don't know" | 5% |

**Table 5.** Middle Leader Definition of Māori Student Success.

| Theme | How Do You Define Māori Student Success? | % ML Comments (n = 170) |
|---|---|---|
| Achieving Success 46.4% | Academic, e.g., grades/NCEA credits | 24.0% |
| | Holistic or general success, e.g., meeting own goals/pride in self/well-being | 19.6% |
| | Student is future-focused | 2.8% |
| Student engagement factors 37.4% | Student feels comfortable in class and school/confident | 13.1% |
| | Student is striving to be best they can be | 5.0% |
| | Student is engaged/enjoying school | 16.2% |
| | Student has positive relationships/leadership capability | 3.1% |
| Cultural factors 10.6% | Student is proud of their culture | 2.2% |
| | Māori students achieving success as Māori | 8.4% |
| No response 5.6% | Blank or "I don't know" | 5.6% |

**Table 6.** Middle Leader Perspectives Regarding the Evidence That Culturally Responsive Practices Make a Positive Difference for Māori Students.

| Theme | Coded Responses | % of ML Comments (n = 170) |
|---|---|---|
| Student experiencing success 29.4% | Academic success or progress/grades | 20.4% |
| | Holistic success | 9% |
| Student engagement 47.1% | Positive attitude/engagement | 23.9% |
| | Improved attendance | 4.2% |
| | Students showing pride in culture/connecting culture to learning | 4.8% |
| | Positive student feedback | 14.2% |
| Positive relationships 15.6% | Student–teacher relationship | 10.7% |
| | Whānau connection/partnership | 4.8% |
| No response 8% | Blank or "I don't know" | 8% |

## 6. Discussion

This study adds to the scant research examining how curriculum middle leaders in secondary schools enact culturally responsive approaches aimed at improving students' academic outcomes. A recent review of the literature investigating middle leadership in New Zealand secondary schools provided no examples of published empirical research investigating middle leaders' support or focus on improving outcomes for Indigenous Māori students [9]. The results of this study provide some insights into how middle leaders theorise and action culturally responsive pedagogies in their leadership roles. The study also sheds light on the ways middle leaders have interpreted and enacted government policy regarding culturally responsive practice in their departments.

The middle leaders in this study defined Māori student success in ways that largely aligned with Ka Hikitia, a government policy designed to raise Māori student achievement [26,47]. Middle leader definitions of Māori student success echoed the good work of government-funded CRP professional development opportunities designed to give effect to Ka Hikitia. Influential research and professional development programmes included Te Kotahitanga [64], which drew attention to the importance of relationships between teachers and students in the classroom; the Starpath Project [41], which emphasised the importance of data utilisation, academic counselling and target setting; He Kākano [65], which focused on culturally responsive leadership development; and Kia Eke Panuku, which drew together the findings from previous projects to provide further professional development opportunities for leaders and teachers in secondary schools [10,66]. These projects all promoted a strategic change-management approach that required school leaders to self-review their evidence of Māori students' participation and achievement, to be open to the views of others, and to make the necessary personal and professional changes at the classroom and school level to ensure Māori students enjoy and achieve educational success as Māori.

Middle leader participants in this study had less clarity regarding the evidence of the culturally responsive practices likely to positively impact Māori students, with just 24% noting that academic achievement was a useful indicator of success. In a recent study of Māori parents' aspirations for their children, the authors found that most whānau (65%) wanted their children to go onto higher education such as polytech or university [67]. In line with these aspirations, research has long focused on better understanding how school leaders might improve opportunities for Māori students to enter degree-level study after secondary school. Key barriers to Māori students attending university include unequal access to relevant NCEA subjects and relevant standards for university pathways [68]; failure of Māori students to reach the literacy standards required for university entrance [69]; and in some schools, a lack of attention to careful curriculum design, flexible timetable structures that enable student subject choice, regular course audits, and guided review of options and opportunity [70]. Our study shows that UE rates for Māori students were highly variable across all of the 12 schools. In order to support Māori parents' aspirations for their children to attend higher education, middle leaders must be more focused on tracking and monitoring the academic outcomes of Māori students. As such, alongside building strong, supportive and positive relationships with Māori students, middle leaders must facilitate an unrelenting departmental focus on data utilisation and analysis for predicting and intervening in achievement conundrums, and provide leadership in finding locally effective solutions. McKinley and Webber also proposed that middle leaders use different forms of data to probe more deeply into departmental structures and dynamics that contribute to inequities in outcomes, or to find alternative approaches to current departmental practices [41]. Although strong teacher–student relationships are critical for Māori student engagement and persistence at school, academic achievement and success must be a central concern for middle leaders in terms of Māori students succeeding "as Māori".

Middle leader participants revealed high levels of overall self-efficacy regarding culturally responsive practice and this appeared unrelated to the socioeconomic status of

the students in the school, or the UE achievement rates for Māori students in their school. However, middle leaders from the three schools with the lowest percentage of Māori students on their rolls had the lowest self-efficacy regarding their culturally responsive practice. In a recent study by the Education Review Office, teachers reported having limited awareness of learners' cultural and learning needs, and more than half of the teachers in the study did not feel confident connecting with ethnic communities [71]. The report contended that, to be more culturally responsive, teachers must seek to understand, value and respect learners' culture in their education, by connecting and partnering with families and communities. In a New Zealand study investigating secondary teachers' beliefs about the relationship between students' cultural identity and their ability to think critically, Davies et al. found 53% of the 490 participants had negative attitudes about the relationship between students' cultural identity and family background and their ability to think critically [72]. With regard to the findings of the present study, the directive for middle leaders to help teachers in their departments more deeply connect and partner with the community seems as important for schools with low numbers of Māori students on the roll, as for those schools with high numbers. It is imperative that we find ways to strengthen the cultural competence and efficacy of curriculum middle leaders to effectively lead their departmental programmes and instruction to enable partnership approaches that support the aspirations of Māori students, whānau and communities.

In this study, only 56% of middle leaders felt confident in providing Māori whānau with opportunities to share their knowledge and experiences in their classrooms, and 31% felt confident teaching the Māori history associated with where their school was based. This result is concerning because integrating Māori knowledge, language, culture and expertise into classroom teaching can be considered a decolonising project [43]. Culturally responsive practice can be improved when middle leaders work alongside teachers in their departments to build strong and enduring connections with Māori students and families, and when whole departments make efforts to learn about and integrate the local history of their school context into Māori students' learning. These efforts enable both Māori students and their teachers to challenge and counter the deliberate and persistent rhetoric of Māori underachievement [43]. Middle leaders have the power, influence and capacity to ensure Māori students are empowered to learn about their own histories and are taught in inclusive classrooms that focus on Māori potential and not deficit.

Only 9% of middle leaders mentioned the use and correct pronunciation of te reo Māori in the classroom as important for Māori student learning. The latest MOE data show increasing numbers of students in New Zealand who are learning te reo Māori in both mainstream schools and Māori-immersion settings [73]. As of 1 July 2022, 27% of the total school population was involved in Māori language in English medium schools, compared to 25.3% in July 2021 [73]. When middle leaders and teachers support te reo Māori by using it within the classroom context, Māori student identity and culture are validated and this promotes the motivation and engagement of Māori students [74]. Newly graduated teachers are expected to have some te reo Māori capability by the time they finish their teaching qualification so that they can normalise its use in classrooms, which is crucial for the vitality of te reo Māori [75]. However, in line with Devine et al. [76], we argue that te reo Māori has to be more than a tick-box exercise for teachers, and te reo Māori use will not alter the negative experiences of Māori students on its own.

The middle leaders in this study had strong theories about supporting Māori student success as Māori, particularly in terms of attending to the pastoral needs of students and nurturing positive teacher–student relationships. However, the strategies they employed to give effect to their theories, and the measures they used to evidence their CRP, were less apparent in their survey responses. International research evidence reveals there is an enactment gap between official policy and practice in schools, or espoused theories of action and the actual theories-in-use, which can negatively impact equal educational opportunity [77,78]. These espoused theories are clearly evidenced in the comments appended in Table 4, and the lack of effective measures for evidencing educational success

is shown in Table 5. Sinnema et al. have argued that attention is rarely given to teacher and leader beliefs about the causes of problems [79].

Maguire et al. argued that the positionality, experience, allegiances and disciplinary commitments of teachers, as well as their loyalties and in-school relations, play out in how policies are interpreted and enacted in multifaceted and complex ways [80]. Cultural constructions of difference and school success and failure are often represented in educators' personal beliefs, attitudes and values and therefore shape how educators interpret CRP as they interact with students. In a study on middle leaders' policy enactment in secondary schools in Ireland, Skerritt et al. described middle leaders as policy translators and noted the inconsistent approach to enactment which could be attributed to being overloaded and inundated with policy work [81]. Maguire et al. [80] warned that not all teachers participate in policy interpretations and translations as their priorities may be elsewhere, and Seashore Louis and Robinson [78] explained that the quality of instructional leadership in a school is likely to be enhanced by the degree of coherence between leaders' personal agendas and the policy agenda. Government policy frameworks designed to address equity issues at a strategic school level will never be realised if middle leaders and teachers have inadequate conversations about how to solve equity problems and effectively measure change. Castagno argued that the "culture of nice" [82] (introduction) in education and our inherent requirement for harmonious workplace relationships mean that middle leaders are likely to struggle with challenging teachers and providing evidence of their shortcomings. As a consequence, middle leaders only give feedback on good, promising and improved practices, rather than the misalignment between departmental goals and actual practice.

In a recent study examining the impact of teachers' high expectations conducted in Western Australian public schools, researchers found teachers communicated high expectations through communicating confidence in students' abilities, adopting engaging and active teaching approaches, developing positive teacher–student relationships, and ensuring an orderly and respectful, emotionally safe learning environment [53]. The students in this Australian study recognised high expectations, and then described how they responded by becoming motivated, engaged in learning and acting to improve their academic outcomes. The majority of middle leaders in the current study identified relationships as key to effective teaching of Māori students. Conversely, fewer than half the participants had high or any expectations for Māori student academic success or mentioned this as an important indicator of Māori student success. In addition, only 2.8% of middle leaders referred to Māori students setting academic goals for themselves or tracking their own academic progress. This study shows that although middle leaders have clearly internalised the messaging of government policy in relation to CRP, they are yet to translate this into equitable practices that result in increased Māori student UE rates.

Two-thirds (8/12) of the secondary schools in this study reported Māori student UE rates that were below the national average in the year of the study. We contend that curriculum middle leaders' relentless focus on building and sustaining positive relationships with Māori students might mean that they have turned their attention away from the need to track and monitor Māori student academic achievement. We argue the evidence in this study reveals middle leaders relieve themselves of the guilt of Māori student underachievement by instead focusing on the centrality of students feeling good about themselves or working at 'their best'. These low expectations are as good as no expectations [53] if they do not translate directly into achievement that will provide a positive platform for Māori students' future aspirations.

## 7. Limitations

This study presents the beliefs and self-reported practices of middle leaders regarding their enactment of CRP within their role as a middle leader and compares these results to whole-school academic results of Māori students. The researchers did not compare practices at the department level where specific Māori student academic scores could be directly correlated with the CRP beliefs of a specific middle leader. There is also no fine-grained

analysis of Māori student academic achievement, rather this study uses the highest level of academic achievement (UE) as the indicator of success. This research highlights general patterns of inconsistency in both CRP practice and academic outcomes for Māori students. Therefore, ongoing research into the knowledge, skills and competencies required by middle leaders to specifically raise Māori student academic achievement will be crucial [9].

**8. Conclusions**

There is a growing awareness of the transformational potential of curriculum middle leaders as academic and instructional leaders in schools. Middle leaders are important conduits for the implementation of key government policies and priorities and can provide essential support and guidance for teachers in their departments [83]. The key to activating this potential is middle leaders aligning their espoused theories with their actual leadership practices. In essence, most demands for culturally responsive, sustaining, revitalising pedagogies and leadership require teachers and leaders to utilise students' culture as a vehicle for learning, but our findings also suggest that middle leaders must be more systematic about monitoring achievement data, and use them to support better learning outcomes for Māori students. While CRP requires middle leaders to help their teachers understand the context in which they teach, question their own knowledge base and assumptions, and build enduring learning relationships with diverse students, we also argue that middle leaders need to maintain a relentless focus on tracking and monitoring Māori student achievement and progress. If the vision of equitable, inclusive education is to be realised, middle leaders will need to lead their teaching teams towards delivering a culturally responsive pedagogical approach and provide opportunities to learn about appropriate tools and measures to test and refine their own learning and teaching theories.

**Author Contributions:** Conceptualization, C.H. and M.W.; methodology, C.H., M.W. and R.W.; software, R.W.; validation, C.H. and M.W.; formal analysis, C.H., M.W. and R.W.; investigation, R.W.; data curation, R.W.; writing—original draft preparation, C.H.; writing—review and editing, C.H., M.W. and R.W.; visualization, R.W.; supervision, C.H.; project administration, R.W.;. All authors have read and agreed to the published version of the manuscript.

**Funding:** This research received no external funding.

**Institutional Review Board Statement:** Ethical approval for the study was granted by the University of Auckland Human Eth ics Committee (UAHPEC Approval Number: 021775) and included voluntary informed consent.

**Informed Consent Statement:** Informed consent was obtained from all subjects involved in the study. Participant confidentiality and anonymity were preserved in the collection, analysis and storage of data.

**Data Availability Statement:** The data is held on a password protected computer owned by the University of Auckland.

**Conflicts of Interest:** The authors declare no conflict of interest.

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
