# Peer review of "Culturally Responsive Middle Leadership for Equitable Student Outcomes"

_education, doi:10.3390/educsci14030327_

Round 1

Reviewer 1 Report

Comments and Suggestions for Authors

This article, titled “Culturally Responsive Middle Leadership for Equitable Student Outcomes” could benefit from some modifications in order to enhance its academic soundness and aid the reader’s understanding.

Even though the topic may be admittedly interesting, the way that the methodology is presented is challenging. The way the statistical analysis was conducted is merely descriptive and cannot lead to safe and comparable results. No comparative research was done in this paper, so the conclusions are not supported by any results. A systematic literature review with similar research designs on the same subject could have been an alternative approach to the same end.

Furthermore, the importance of this research and its academic contribution is not highlighted anywhere. What research gap does it come to fill in and why is this important? Wherein lies its originality vis-a-vis past research on the same field or how is this originality substantiated in any other fashion? Moreover, the limitations of the present research are not clearly defined, nor are suggestions for future research mentioned in the end that would help frame the insights of this research design.

In fact, a lack of an up-to-date bibliographical references is observed throughout the whole manuscript and especially in the section of introduction. Further bibliographic reference would be appropriate in the introduction, to properly frame and solidify the research design, its questions and overall aim while an updated bibliography would be recommended in lines 47-60.

Some more specific observations that underline this point follow. For instance, line 25 needs an in-text citation while line 28 could use rephrasing, it is a research question. It must be clearly stated. In line 36, do you have another research question? Where are the attitudes and values mentioned in line 26?

It is important to have a clear statement of the purpose of the present research followed by the research questions. Also, it should be clearly stated which variables are being studied and a consistency should be maintained in their reference that corresponds with the presentation of the results.

Again, in line 103-104, above had other variables (line 26) such as values and attitudes which are now not mentioned. After all, what are the variables to be studied and in relation to what? Likewise, the research instrument mentioned in line 219, is designed to study what variables? What are its proper psychometric characteristics? Has there any exploratory or confirmatory factor analysis taken place?

Last but not least, there is certainly need for proper formatting according to the journal’s instructions and for an improvement altogether of the academic language throughout the paper, although this can be also done at a later time. What is of utmost importance is primarily to solidify the research design and to that aim specific parts of the paper need extensive revision.

I sincerely hope the aforementioned comments can help in this direction.

Best regards & Season’s Greetings,

Reviewer

Author Response

Please see word doc attached

Reviewer 2 Report

Comments and Suggestions for Authors

The article addresses a relevant issue in the current educational context. Overall, it is well structured and supported by the literature on the subject. The theoretical framework is clear, although it does not delve into the concept of leadership and its relationship with the concept of management. An analysis of the theoretical and conceptual differences between leadership and management could help to deepen the different dimensions of middle leadership. On the other hand, considering the international context, it would be useful to briefly contextualise the political system of school governance in New Zealand (more or less centralised), in order to understand the degree of autonomy of schools and in particular leadership in the development of Equitable Student Outcomes. This information would enrich the intersectional analysis proposed by the authors, by making the articulation between the structural, political and representational levels more explicit.

Author Response

Memo in response to reviewers comments attached

Round 2

Reviewer 1 Report

Comments and Suggestions for Authors

The writers have implemented the suggested revisions.

Author Response

The reviewer states that the writers have implemented the suggested revisions